# Head-to-Head Comparison of [^18^F]PSMA-1007 and [^18^F]FDG PET/CT in Patients with Triple-Negative Breast Cancer

**DOI:** 10.3390/cancers16030667

**Published:** 2024-02-04

**Authors:** Natalia Andryszak, Daria Świniuch, Elżbieta Wójcik, Rodryg Ramlau, Marek Ruchała, Rafał Czepczyński

**Affiliations:** 1Department of Endocrinology, Metabolism and Internal Medicine, Poznan University of Medical Sciences, 61-701 Poznan, Poland; mruchala@ump.edu.pl (M.R.); czepczynski@ump.edu.pl (R.C.); 2Department of Nuclear Medicine, Affidea, 61-485 Poznan, Poland; 3Department of Oncology, Poznan University of Medical Sciences, 61-701 Poznan, Poland; 4Department of Oncology Medical Center HCP Poznan, 61-485 Poznan, Poland

**Keywords:** triple-negative breast cancer, PSMA, PET/CT

## Abstract

**Simple Summary:**

This study investigates a promising avenue for improving the diagnosis and treatment of triple-negative breast cancer (TNBC), a highly aggressive form of breast cancer with limited therapeutic options. The researchers focus on the prostate-specific membrane antigen (PSMA), known for its presence in prostate cancer but also identified in breast cancer. Using ^18^F-PSMA-1007 PET/CT, the study aims to assess PSMA’s in vivo expression in TNBC patients and compare it with the standard [^18^F]FDG PET/CT. The findings suggest that [^18^F]PSMA-1007 PET/CT may outperform current methods in detecting distant metastases, especially in the brain. This research not only enhances our understanding of PSMA expression in TNBC but also hints at potential advancements in diagnostic imaging and targeted therapies for this challenging cancer type.

**Abstract:**

Background: Triple-negative breast cancer (TNBC) exhibits high aggressiveness and a notably poorer prognosis at advanced stages. Nuclear medicine offers new possibilities, not only for diagnosis but also potentially promising therapeutic strategies. This prospective study explores the potential of prostate-specific membrane antigen (PSMA) as a diagnostic and therapeutic target in TNBC. Methods: the research investigates PSMA expression in vivo among TNBC patients using [^18^F]PSMA-1007 PET/CT and compares it head-to-head with the standard-of-care [^18^F]FDG PET/CT. Results: The study involves 10 TNBC patients, revealing comparable uptake of [^18^F]PSMA-1007 and [^18^F]FDG in primary and metastatic lesions. Nodal metastases were found in eight patients, showing similar SUV_max_ values in both modalities. Two patients had uncountable lung metastases positive in both [^18^F]FDG and [^18^F]PSMA-1007 scans. PET-positive bone metastases were identified by ^18^F-PSMA in four patients, while elevated [^18^F]FDG uptake was found only in three of them. Distant metastases displayed higher SUV_max_ values in the [^18^F]PSMA-1007 PET/CT, as compared to [^18^F]FDG. Additionally, brain metastases were exclusively detected using [^18^F]PSMA-1007. Conclusions: the findings provide valuable insights into the expression of PSMA in TNBC and underscore the potential clinical significance of [^18^F]PSMA-1007 PET/CT in enhancing both diagnostic and therapeutic approaches for this aggressive breast cancer subtype.

## 1. Introduction

Triple-negative breast cancer (TNBC) accounts for approximately 20% of breast cancer cases, commonly affecting younger patients (under 40 years old). This subtype is characterized by higher aggressiveness and a greater tendency to generate metastases. Although long-term survival rates for stage I TNBC are similar to other subtypes, the prognosis significantly worsens with advanced stages [1,2]. Nearly half of all women diagnosed with stage III disease die within four years of diagnosis, primarily due to treatment resistance and the absence of alternative therapeutic strategies [1]. Due to the lack of estrogen and progesterone receptor expression, as well as the low expression of the HER2 epidermal growth factor receptor, TNBC is characterized by primary resistance to hormone therapy and insensitivity to immunotherapy using trastuzumab and pertuzumab. The significant heterogeneity of cancer cells and a high proliferation rate worsen the prognosis due to the high potential for distant metastasis. Treatment relies on multi-drug chemotherapy (CTH) regimens and recently introduced checkpoint inhibitors [2]. Unfortunately, recurrence resistance to treatment is very common. Research into new treatment methods does not rely solely on systemic therapies. Nuclear medicine could be useful in TNBC patients for both diagnostic and therapeutic purposes.

PET/CT examination using ^18^F-fluorodeoxyglucose ([^18^F]FDG) is mainly used in TNBC patients to assess recurrence after radical treatment when other imaging modalities are inconclusive, less often in staging before treatment [2]. Nevertheless, the significant development in nuclear medicine, concerning both the availability of radiopharmaceuticals and new procedures, leads to an expanded scope of its utilization. The definition of prostate-specific membrane antigen (PSMA) as a target for radionuclide diagnosis and therapy of prostate cancer has become a clinical breakthrough in nuclear medicine. PET/CT using PSMA ligands labeled with gallium-68 (e.g., [^68^Ga]Ga-PSMA-11) or with fluorine-18 (e.g., [^18^F]PSMA-1007) has become a standard diagnostic tool [3], and more recently, PSMA ligands labeled with beta or alpha radiation emitters have been introduced into the treatment of disseminated prostate cancer. The analysis of the effectiveness of targeted radionuclide therapy using ^177^Lu-PSMA-617 in nearly 150 patients showed high response rates and safety [3,4].

PSMA is a glycoprotein present in the prostate, primarily in the cytoplasm and the upper layers of the glandular ductal epithelium. In the process of malignant transformation, a significant overexpression of PSMA occurs together with a displacement of the antigen’s location toward the glandular lumens [5]. In as early as 1999, Chang et al. immunohistochemically proved the presence of PSMA in other cancer cells, including the endothelium of kidney cancer, bladder, melanoma, testicular cancer, and breast cancer. In prostate cancer, increased levels of PSMA directly correlate with greater aggressiveness [6]. A probable role of PSMA in tumor neoangiogenesis has been demonstrated. Importantly, its expression has not been found in non-cancerous tissues. Within breast cancer, Tolkach et al. confirmed PSMA expression in 60% of patients [7]. The highest expression was observed in triple-negative breast cancer, where PSMA presence was confirmed in all 33 patients.

The previous utilization of PSMA expression in the radionuclide treatment of prostate cancer, as well as the first published observations regarding the presence of this antigen on breast cancer cells, encourage similar clinical studies concerning the therapy of breast cancer in the future. Due to the likely highest expression of the PMSA antigen, it appears that TNBC could be the optimal diagnosis for the treatment with radiolabeled PSMA ligands. Recently, results from the first clinical study using the labeled PSMA ligand-^177^Lu-PSMA in one patient with triple-negative breast cancer have been published [7]. Due to the advanced stage of the disease and rapid progression, the patient did not complete the treatment, preventing an evaluation of the therapy's effectiveness. The confirmation of PSMA antigen presence in TNBC cells and neovasculature serves as a rationale for the more extensive use of radioactively labeled PSMA ligands in PET imaging and individualized radionuclide therapy in this aggressive form of breast malignancy.

The aim of our study was to assess the expression of PSMA in vivo in TNBC patients by means of PET/CT using [^18^F]PSMA-1007 and to compare it to the standard-of-care [^18^F]FDG PET/CT in TNBC patients.

## 2. Materials and Methods

This prospective study was approved by the Institutional Ethics Committee.

Between March 2023 and September 2023, 10 patients with triple-negative breast cancer in different stages of the disease (10 women; mean age 59.3; range 38–72 years) were prospectively enrolled in the study. Patients were referred for [^18^F]FDG PET/CT for different indications: to obtain staging of the disease before treatment, to evaluate treatment effect, or to confirm TNBC recurrence after the treatment. Written informed consent was obtained from each patient.

Detailed information on demographics and clinical history, including treatment history, tumor site, and detection methods, was recorded. Then, each patient underwent [^18^F]PSMA-1007 PET/CT examination, with no more than one month interval between the scans.

### 2.1. PET/CT Examinations

Whole-body PET/CT examinations were performed using a Discovery IT scanner (GE Healthcare, Chicago, IL, USA) equipped with a 64-slice computed tomography. The images were acquired 60 min after injection of [^18^F]FDG (Glunektik by Synektik, Warszawa, Poland) or [^18^F]PSMA-1007 (also manufactured by Synektik, Poland) with an activity of 3 MBq/kg body weight for each examination. For the sake of clarity of the text, an abbreviated form of the radiopharmaceutical, [^18^F]PSMA-1007, will be used.

The examination covered the standard whole-body range, from the top of the head to the mid-thigh, with the patient lying on their back with arms raised. Patients underwent a low-dose computed tomography (CT) prior to PET-processed acquisition for attenuation correction and anatomical correlation of PET findings. The CT images were obtained using the following parameters: 1.25 mm layer thickness, 1.375:1 pitch, 50 cm DFOV 50, and 512 × 512 matrix. No contrast enhancement was used for the CT imaging. PET/CT exams were performed with a standard acquisition time of 1.5 min per bed position. Emission data were corrected for randoms, dead time, scatter, and attenuation and were reconstructed using the Q.Clear algorithm to obtain optimal lesion delineation.

### 2.2. Image Analysis

The images were analyzed with the use of Advantage Workstation (GE Healthcare). All scans were interpreted separately by two experienced nuclear medicine specialists. Both [^18^F]PSMA-1007 and [^18^F]FDG PET/CT results were compared. PET, CT, and fused PET/CT images were reviewed. PET images were first assessed visually, using transaxial, sagittal, and coronal displays. For the quantitative analysis, the maximal standard uptake value (SUV_max_) of each positive lesion was measured on both PET/CT images. The SUV_max_ measurement was performed using spherical volumes of interest (VOIs) with a diameter adapted to lesion size. Target-to-background ratios (TBRs) were calculated using the SUV max of the lesion divided by the SUV max of the background that was measured using a volume of interest of a similar diameter placed in an unaffected region.

## 3. Results

Ten patients with TNBC were included in the study: an untreated patient with a newly diagnosed TNBC (#1), a patient after neoadjuvant CTH (#2), five patients undergoing palliative CTH (#3–7), and three patients with a suspicion of TNBC recurrence after treatment (#8–10) (Table 1). Five patients had been treated with mastectomy (#6–10). At the time of recruitment, four patients were not undergoing any oncological treatment (#6, #8–10), and the time from the last course of CTH ranged from 2 to 45 months (median 13 months). Six other patients were under continuous CTH as the active disease had been diagnosed with other imaging modalities (computed tomography, magnetic resonance, and ultrasonography). Findings obtained in both PET/CT modalities are summarized in Table 2.

**Table 1 cancers-16-00667-t001:** Diagnosis and treatment history of the patients. CTH—chemotherapy, RTH—radiotherapy, DOX—doxorubicin, CPA—cyclophosphamide.

Patient	Age	Diagnosis	History of Treatment	Timing of PET and Current Treatment
#1	38	diagnosed August 2023	no treatment	staging September 2023
#2	54	diagnosed March 2022 T4bN0M1, Ki67 80%	after 12 DOX + CPA, during docetaxel, 3rd month	neoadjuvant CTH February 2023
#3	72	diagnosed October 2022, T2N1M1, KI67 90%,	during palliative CTH, 5th month (DOX + CPA)	palliative CTH June 2023
#4	48	diagnosed November 2022, T3N1M1, Ki67 30%, BRCA 1 and 2 mutation	during palliative CTH, 4th month (DOX + CPA)	palliative CTH February 2023
#5	66	diagnosed March 2022, T2N3aM1, Ki67 50%,	during palliative CTH, 12th month (capecitabine)	palliative CTH June 2023
#6	57	diagnosed January 2022, T2N0M0	neoadjuvant CTH January 2023 mastectomy+lymphadenectomy adjuvant CTH (capecitabine) completed August 2023	finished treatment, suspicion of recurrence September 2023
#7	63	diagnosed 2021 Ki67 90%	2021–mastectomy adjuvant CTH and RTH September 2022–recurrence during palliative CTH, 6th month, DOX + paclitaxel	palliative CTH February 2023
#8	61	diagnosed 2021, Ki67 > 50%	2021–neoadjuvant CTH, mastectomy + lymphadenectomy, adjuvant CTH (capecitabine) and RTH completed December 2022	finished treatment, suspicion of recurrence, September 2023
#9	72	diagnosed 2019, Ki67 20%, positive BRCA 1 and 2, CHEK2 and PALB2 mutations	2019–mastectomy adjuvant CTH (DOX + paclitaxel) completed in 2019	finished treatment, suspicion of recurrence, June 2023
#10	62	diagnosed 2017, Ki67 60%	2017–neoadjuvant CTH, mastectomy, and adjuvant CTH 2019–recurrence–CTH(capecitabine + paclitaxel) January 2023–progression during palliative CTH (carboplatin)	palliative CTH February 2023

**Table 2 cancers-16-00667-t002:** Summary of the PET/CT findings obtained with [^18^F]FDG and [^18^F]PSMA-1007. Abbreviations: FDG—PET/CT using [^18^F]FDG; PSMA—PET/CT using [^18^F]PSMA-1007; LN—lymph node; mets—metastasis; SUV_max_—maximal standardized uptake value, TBR—target-to-background ratios.

Patient	PET/CT Using [^18^F]FDG Findings	SUVmax [^18^F]FDG	TBR [^18^F]FDG	PET/CT Using [^18^F]PSMA-1007 Findings	SUV_max_ [^18^F]PSMA-1007	TBR [^18^F]PSMA-1007	Comparison of Both Modalities
#1	1 breast lesion	3.4	2.8	2 breast lesions	6.0	2.8	one breast lesion more in PSMA (Figure 1)
#2	1 axillary LN	1.2	3.0	1 axillary LN	1.2	4.0	similar SUV_max_
#3	breast tumor, hilar LN, subcarinal LN	4.0 7.0 3.9	5.7 7.0 2.8	breast tumor, hilar LN	2.4 6.2	4.0 6.9	SUV_max_ higher in FDG, 1 LN more in FDG
#4	left breast tumor, right breast lesion, axillary LN, lung mets, liver mets, adrenal mets, bone mets	4.0 4.5 4.5 2.3 4.1 2.3 5.3	2.5 4.5 9.0 3.3 1.8 1.7 7.6	left breast tumor right breast lesion, axillary LN, lung mets, liver mets, adrenal mets, bone mets, brain mets	4.5 1.1 9.6 5.4 17.3 11.7 9.8 5.9	5.0 1.0 32.0 5.9 3.4 2.3 8.2 59.0	right breast lesion with higher SUV_max_ in FDG, SUV_max_ values much higher in distant mets in PSMA, brain mets detectable only in PSMA (Figure 2 and Figure 3)
#5	breast tumor, axillary and mediastinal LN, lung mets, bone mets	3.5 3.7 3.8 3.8	8.7 2.8 9.5 3.2	breast tumor, axillary and mediastinal LN, lung mets, bone mets	2.1 3.2 2.4 6.8	4.2 2.0 4.8 5.7	equal number of detected lesions, higher SUV_max_ in FDG, except a bone lesion with SUV_max_ higher in PSMA
#6	no lesions detected	-	-	no lesions detected	-	-	Complete metabolic response
#7	axillary LN, mediastinal LN, lung mets	3.8 3.8 12.0	5.4 2.3 20.0	axillary LN, lung met	4.7 3.1	11.7 4.4	SUV_max_ higher in FDG (lung mets) or PSMA (LN mets), 1 LN more in FDG (Figure 4)
#8	local recurrence, regional LN, hilar LN, pleural mets, bone mets	3.9 5.2 5.6 6.1 16.7	4.3 4.6 4.7 7.6 11.9	local recurrence, regional LN, hilar LN, mediastinal mets, pleural mets, bone mets	2.5 1.6 1.9 2.4 3.3 18.1	2.5 2.3 2.1 2.4 4.1 15.0	SUV_max_ higher in FDG (local recurrence and LN) or PSMA (bone mets), 1 LN more in PSMA, more bone lesions in PSMA (especially in cranium)
#9	mediastinal LN	3.3	2.1	mediastinal LN, 2 bone mets	2.9 6.2	2.4 6.9	one mediastinal LN more in PSMA, bone mets only in PSMA (Figure 5)
#10	mediastinal LN, abdominal LN	4.3 2.9	2.5 1.5	mediastinal LN, abdominal LN, 1 bone mets	4.4 3.6 7.3	2.3 2.6 6.1	similar SUV_max_ bone mets only in PSMA (Figure 6)

**Figure 1 cancers-16-00667-f001:**
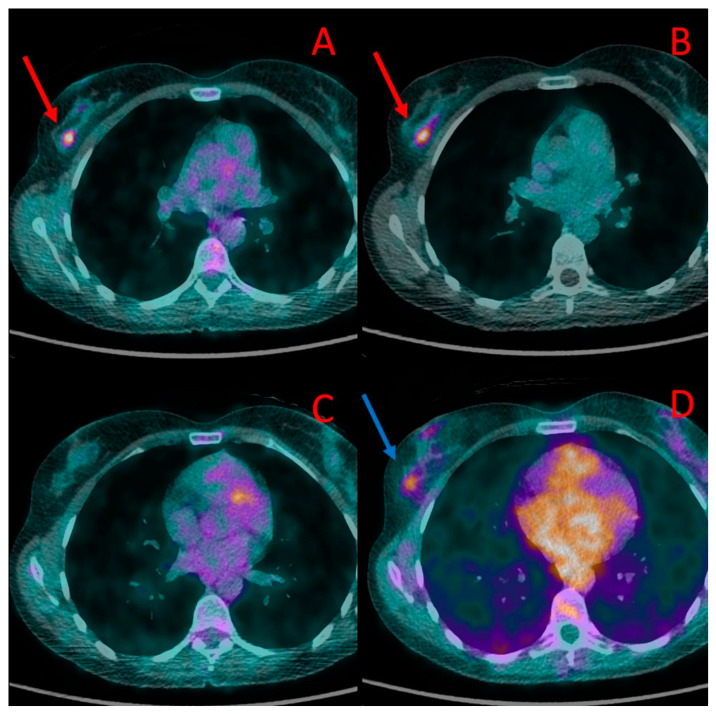
Primary breast lesions in the right breast depicted in PET/CT in patient #1. The first lesion is visible in PET/CT performed with [^18^F]FDG (**A**) and [^18^F]PSMA-1007 (red arrow) (**B**). A second lesion, located slightly below, was visible in [^18^F]PSMA-1007 (blue arrow) (**D**) but not with [^18^F]FDG (**C**). Both lesions were histologically confirmed as TNBC.

**Figure 2 cancers-16-00667-f002:**
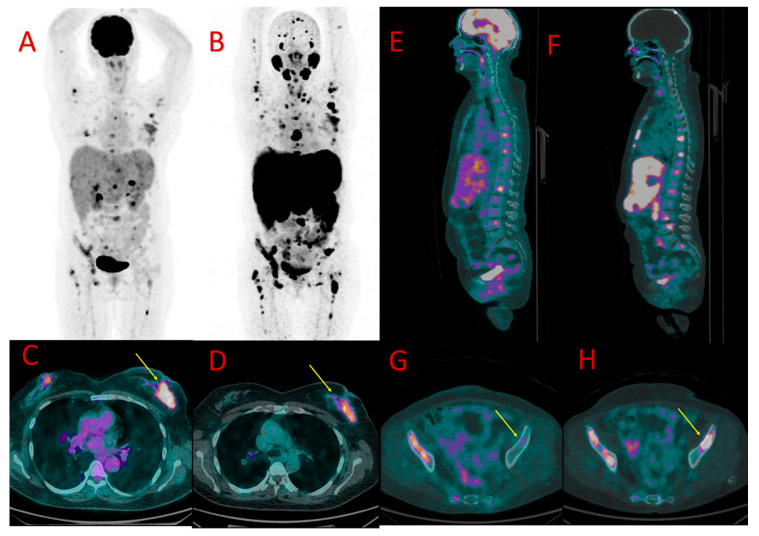
[^18^F]FDG (**A**) and [^18^F]PSMA-1007 (**B**) PET in patient #4 showing the primary breast tumor, axillary lymph node metastases, and dissemination of the TNBC to the bones, liver, and lungs. Similar uptake of [^18^F]FDG (**C**) and [^18^F]PSMA-1007 (**D**) was found in the primary lesion in the left breast (SUV_max_ 4.0 vs. 4.5, TBR 2.5 vs. 5.0, respectively), arrows. More metastatic bone lesions in vertebrae (**E**,**F**) and in the pelvis ((**G**,**H**), arrows) were found in [^18^F]PSMA-1007 PET/CT (**F**,**H**) than in [^18^F]FDG PET/CT (**E**,**G**).

**Figure 3 cancers-16-00667-f003:**
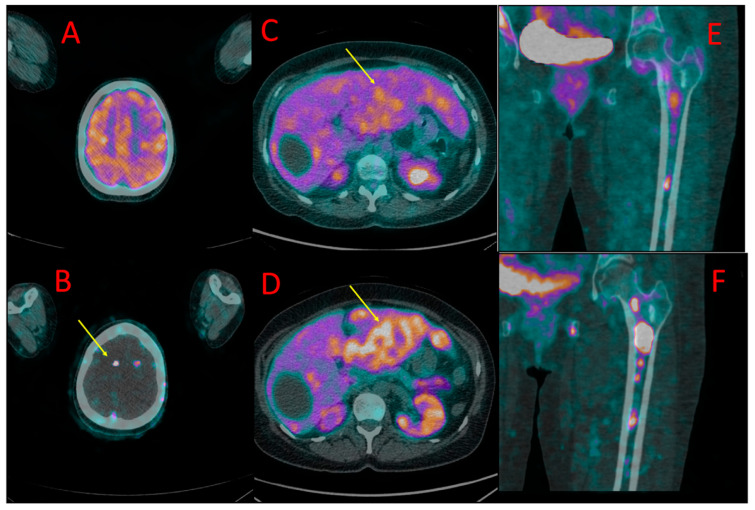
One of the brain metastases in patient #4 detected in [^18^F]PSMA-1007 (**B**) but not in [^18^F]FDG PET/CT (**A**). Diffuse liver metastases showed higher uptake of [^18^F]PSMA-1007 (**D**) than of [^18^F]FDG (**C**). More bone metastases in the left femoral bone were detected, and higher radiotracer accumulation in these lesions was found in ^18^F-PSMA PET/CT (**F**) in comparison to [^18^F]FDG (**E**).

**Figure 4 cancers-16-00667-f004:**
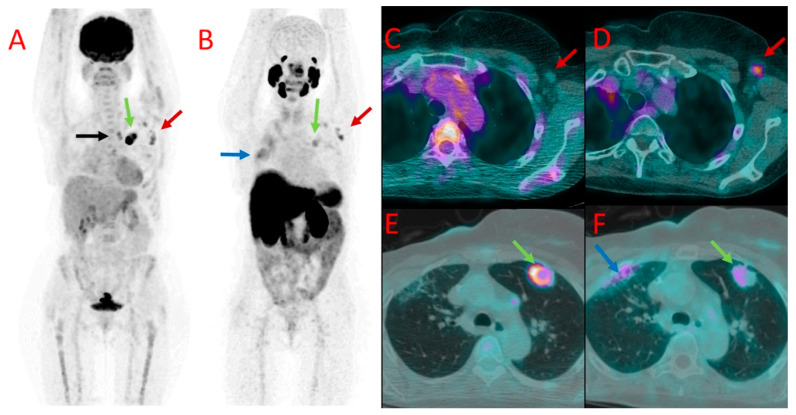
[^18^F]FDG (**A**) and ^18^F-PSMA (**B**) PET as well as [^18^F]FDG (**C**,**E**) and [^18^F]PSMA-1007 (**D**,**F**) PET/CT fused images of the chest in patient #7. Metastatic axillary lymph nodes showed uptake of both tracers (red arrows at (**A**–**D**)). However, mediastinal lymph nodes presented accumulation of [^18^F]FDG only (black arrow at (**A**)). A large metastasis in the left lung (green arrows) also showed a higher accumulation of [^18^F]FDG (**A**,**E**) than [^18^F]PSMA-1007 (**B**,**F**) with the SUV_max_ of 12.0 vs. 3.1, TBR 20.0 vs. 4.4. Benign post-irradiation pulmonary lesions in the right lung showed diffuse uptake of [^18^F]PSMA-1007 only (blue arrow at (**B**,**F**)).

**Figure 5 cancers-16-00667-f005:**
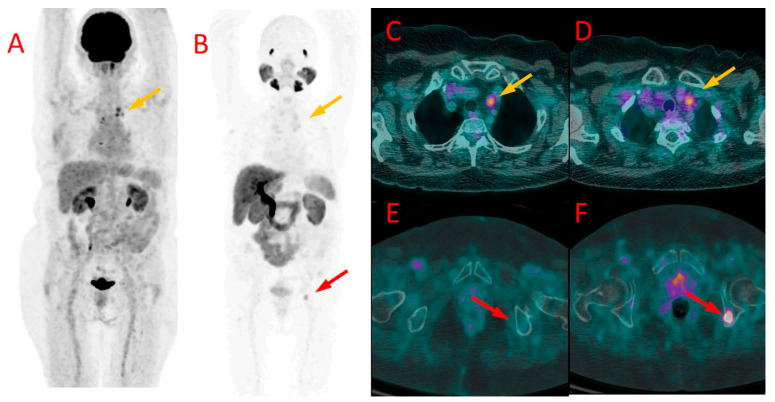
[^18^F]FDG (**A**) and [^18^F]PSMA-1007 (**B**) PET as well as [^18^F]FDG (**C**,**E**) and [^18^F]PSMA-1007 (**D**,**F**) PET/CT fused images of the chest (**C**,**D**) and pelvis (**E**,**F**) in patient #9. Metastatic mediastinal lymph nodes (yellow arrows at (**A**–**D**)) showed uptake of both tracers, slightly higher in [^18^F]FDG scans (SUV_max_ 3.3 vs. 2.9, TBR 2.1 vs. 2.4). On the contrary, bone metastasis in the left iliac bone was detectable only with [^18^F]PSMA-1007 PET/CT (red arrow at (**B**,**E**,**F**)) with SUV_max_ of 6.2 and TBR 6.9, without any osteolytic lesion in the CT (**E**).

**Figure 6 cancers-16-00667-f006:**
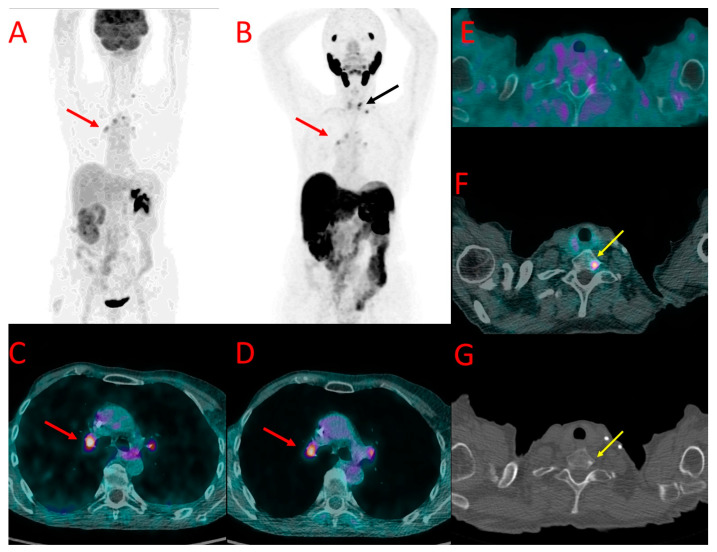
[^18^F]FDG (**A**) and [^18^F]PSMA-1007 (**B**) PET as well as [^18^F]FDG (**C**,**E**) and [^18^F]PSMA-1007 (**D**,**F**) PET/CT fused images of the chest (**C**,**D**) and neck (**E**,**F**) as well as CT of the neck (**G**) in patient #10. Metastatic hilar lymph nodes (red arrows at (**A**–**D**)) showed similar uptake of both tracers. On the contrary, bone metastasis in the Th1 vertebra was detectable only with [^18^F]PSMA-1007 PET/CT (black arrow at (**B**), yellow arrow at (**F**)) with SUV_max_ of 7.3 and TBR 6.1, with a corresponding osteosclerotic lesion in the CT (yellow arrow (**G)**).

### 3.1. Local Disease

In the only treatment-naive patient (#1), both PET/CT modalities showed increased radiopharmaceutical uptake in the tumor of the right breast. The tumor had a diameter of 10–12 mm and showed a higher uptake of [^18^F]PSMA-1007 (SUV_max_ 6.0, TBR 2.8) than [^18^F]FDG (SUV_max_ 3.4, TBR 2.8). In addition, the ^18^F-PSMA PET/CT showed another focus of radiotracer accumulation in the same breast that was not present in the [^18^F]FDG PET/CT. The second focus was verified as a second focus of TNBC (Figure 1).

In four other patients (#3, 4, 5, and 8) undergoing CTH, breast tumor showing accumulation of both tracers was depicted. In three of them, the uptake of [^18^F]FDG was higher than of [^18^F]PSMA-1007, whereas, in the fourth patient (#4), the radiopharmaceutical accumulation was similar (Figure 2C,D).

### 3.2. Nodal Metastases

PET-positive lymph nodes typical for nodal metastases were present in eight patients. In two of them (#2 and 4), PET-positive nodes were restricted to the axillary region. In two other patients (#5 and 7), such lymph nodes were located both in the axillary fossa and in the mediastinum. In three other patients (#3, 8, and 9), metastatic lymph nodes were present in the mediastinum only and in the remaining one (#10)-in both mediastinum and abdomen.

The number of PET-positive lymph nodes, defined as nodes with an uptake higher than the background, was equal in both modalities in 5 patients (#2, 4, 5, 8, and 10). In one patient (#9), [^18^F]PSMA-1007 PET/CT showed one more positive lymph node than [^18^F]FDG. In two pts. (#3, 7) [^18^F]PSMA-1007 PET/CT presented fewer positive nodes than [^18^F]FDG.

The mean value of the highest SUV_max_ in the positive nodes was 4.3 ± 2.7 (range 1.2–9.6) with ^18^F-PSMA and 4.2 ± 1.7 (range 1.2–7.0) with [^18^F]FDG (p-ns).

In six patients with multiple PET-positive nodes, variable values of SUV_max_ were found among individual nodes in both studied modalities. Moreover, no clear relation was found between radiopharmaceutical uptake levels in the nodes in individual cases, e.g., higher ^18^F-PSMA uptake than [^18^F]FDG was observed in some lymph nodes, while other nodes showed higher [^18^F]FDG accumulation than [^18^F]PSMA-1007 in a patient (Figure 4).

### 3.3. Lung Metastases

Two patients (#4, 5) had uncountable lung metastases with maximal diameters of 16 mm and 20 mm. Larger metastatic lesions (above 5 mm) showed the uptake of both radiopharmaceuticals. In one of the patients (#4), higher SUV_max_ values were obtained with [^18^F]PSMA-1007 (SUV_max_ = 5.4 vs. 2.3 for [^18^F]FDG, TBR 5.9 and 3.3, respectively), in the other (#5) with [^18^F]FDG (SUV_max_ = 3.8 vs. 2.4 for [^18^F]PSMA-1007, TBR 9.5 and 4.8, respectively).

In a third patient (#7), a large pulmonary metastatic nodule was found in the left lung. It showed much higher [^18^F]FDG uptake as well (SUV_max_ 12.0 vs. 3.1 for [^18^F]PSMA-1007, TBR 20.0 and 4.4, respectively) (Figure 4E,F).

In a third patient (#8), a solitary pleural metastatic lesion with a diameter of 10 mm was detectable in both modalities. However, a higher SUV_max_ value was obtained with [^18^F]FDG (6.1 vs. 3.3 for [^18^F]PSMA-1007, TBR 7.6 and 4.1, respectively).

### 3.4. Bone Metastases

Altogether, bone metastases were found in five patients (#4, 5, 8, 9, and 10). PET-positive bone lesions characteristic of bone metastases were found in both PET/CT modalities in three patients (#4, 5, and 8). In all of them, the SUV_max_ values measured in the bone lesions were higher in PET/CT with [^18^F]PSMA-1007 (6.8, 9.8, 18.1) than with [^18^F]FDG (3.8, 5.3, 16.7, respectively), with TBR in [^18^F]PSMA-1007 5.7, 8.2, 15.0 and in [^18^F]FDG 3.2, 7.6, 11.9, respectively (Figure 2 and Figure 3E,F). The number of bone metastases ranged from one lesion visible in both methods (#5) to uncountable lesions in two remaining patients (#4, 8), However, the number of [^18^F]PSMA-1007-positive lesions was higher than [^18^F]FDG -positive metastases in patient #8 (in particular, higher detectability of [^18^F]PSMA-1007 PET/CT was observed in cranial bones).

In two other patients (#9 and #10), foci of increased [^18^F]PSMA-1007 uptake were detected in the bones that were not visible in [^18^F]FDG PET/CT. In patient #9, two lesions were found in the iliac bone and Th9 vertebral body (Figure 5). These lesions did not show any morphological correlation in the bone structure; in spite of that, they were interpreted as bone metastases due to their presentation, anatomical location, and relatively high SUV_max_ (6.2, TBR 6.9). In patient #10, a solitary focus of increased [^18^F]PSMA-1007 accumulation (SUV_max_ 7.3, TBR 6.1) was found in the Th1 vertebral body. The lesion was osteosclerotic in the CT and had a diameter of 6 mm. It did not show any [^18^F]FDG uptake (Figure 6).

### 3.5. Distant Metastases to Other Organs

In addition to metastatic lesions in the lymph nodes, lungs, and bones, one of the patients (#4) also developed metastases in the liver, adrenal glands, and brain.

Liver metastases (up to 67 mm) showed higher [^18^F]PSMA-1007 uptake than [^18^F]FDG (SUV_max_ = 17.3 vs. 4.1, TBR 3.4 vs. 1.8) (Figure 3C,D).

Metastases in both adrenals (20 and 55 mm) also presented with higher [^18^F]PSMA-1007 accumulation than ^18^F-FDG (SUV_max_ = 11.7 vs. 2.3, TBR 2, 3 vs. 1.7).

Approximately ten small brain metastases (diameter ranging from 4 to 7 mm) were detected only with the use of [^18^F]PSMA-1007 PET/CT (they were [^18^F]FDG-negative) (Figure 3A,B). The central nervous system metastases were subsequently confirmed using brain MR.

### 3.6. Additional Findings

In patient #3, a thyroid incidentaloma was found in both modalities. The thyroid nodule had a diameter of 22 mm and similar SUV_max_ (2.9 at [^18^F]FDG and 2.7 by ^18^F-PSMA). In addition, a metabolically active lesion measuring 5 mm in the parotid salivary gland was found in the [^18^F]FDG PET/CT only (SUV_max_ 5.4) in that patient. Both thyroid and parotid lesions were verified using ultrasonography and biopsy; they represented a benign thyroid nodule and Warthin’s tumor of the parotid.

In patient #7, thickening of the rectal wall was found, showing elevated uptake of [^18^F]FDG (SUV_max_ 17.7) and only slightly elevated uptake of [^18^F]PSMA-1007 (SUV_max_ 3.0). The lesion was verified using colonoscopy as a benign lesion (tubular adenoma of the rectum).

In patient #10, infiltration of the ureter was found by both modalities with similar SUV_max_ values (3.5–3.6). The infiltration was located in the region of the division of the right common iliac artery. The lesion leading to the dilation of the ureter and hydronephrosis had been found prior to the PET/CT imaging, and it was clinically regarded as a metastasis. Hydronephrosis was treated with a transcutaneous nephrostomy.

## 4. Discussion

Triple-negative breast cancer is characterized by low differentiation, high aggressiveness, and a propensity for distant metastases, contributing to poor prognosis and a high recurrence rate [8]. Due to the absence of estrogen, progesterone, and human epidermal growth factor receptors, the treatment options for TNBC patients are limited. Numerous studies are underway to explore the potential role of the TNBC microenvironment and to identify molecular subtypes that would benefit from effective immunotherapy [8,9]. However, the results are not yet satisfactory, and further studies are necessary to develop novel and precisely targeted treatments for TNBC.

Prostate-specific membrane antigen (PSMA) is a transmembrane glycoprotein primarily recognized for its overexpression in prostate cancer. Its use as a molecular target for radionuclide imaging and therapy has significantly influenced prostate cancer management in recent years. Along with an increasing number of studies highlighting the expression of PSMA in various solid tumors, growing interest in exploring its potential utilization in breast cancer imaging or treatment can be observed. To this day, several studies have evaluated immunohistochemical expression of PSMA in breast cancer [10,11,12,13].

In 1999, Chang et al. first identified PSMA expression not only in prostate cancer but also in renal cancer, colonic cancer, non-small cell lung carcinoma, and breast cancer [6]. In 2004 Ross et al. demonstrated the overexpression of PSMA in tumor-associated vessels in breast cancer [14]. Wernicke et al. subsequently discovered the overexpression of PSMA not only in primary breast cancer cells but also in distant metastases and indicated significantly higher PSMA expression in metastases [11]. Tolkach et al. conducted a comprehensive assessment of PSMA in over 300 cases of breast cancer, revealing a significant correlation between the histological type of breast cancer and PSMA expression [7]. The exact role of PSMA in neoplasia and neoangiogenesis remains unclear. However, studies have indicated that more aggressive tumors with lower differentiation and higher grades exhibit significantly higher PSMA expression. In particular, TNBC has demonstrated the highest PSMA expression.

As an integral membrane cell surface protein, PSMA serves as an excellent target for radionuclide ligands, which have been successfully applied in medical imaging. PET examinations utilizing PSMA ligands labeled with ^68^Ga or ^18^F, both highly effective nuclear imaging tracers, offer the capability to analyze the distribution of cancer cells in the body, particularly in prostate cancer. As a consequence of its remarkable sensitivity, PET/CT using ^68^Ga-labeled PSMA ligands has become a routine tool for staging and detecting recurrence in prostate cancer.

The interest of researchers focused on PSMA as a molecular target extends beyond imaging as the recent introduction of PSMA ligands to radionuclide therapy of prostate cancer has shown beneficial results [3]. An analysis of the effectiveness of radionuclide therapy, specifically using ^177^Lu-PSMA-617, in nearly 150 patients with prostate cancer revealed high response rates and safety. Additionally, its combination with androgen receptor antagonists proved effective in 165 patients with metastatic castration-resistant prostate cancer [4]. In another trial, [^177^Lu]Lu-PSMA-617 appeared superior to cabazitaxel in terms of PSA response [15]. Finally, based on the outcomes of a multicenter trial involving 831 patients, this treatment has been approved in the US and in several other countries [7].

Few studies have reported the results of PET/CT with radiolabeled PSMA ligands in breast cancer patients [12,16,17,18,19], most of which were single-case presentations. Two larger studies are of special interest. Sathekge et al. demonstrated a total of 19 cases (subgroups of PR-positive and PR-negative breast cancer) who underwent ^68^Ga-PSMA-HBED-CC PET/CT with an overall detection rate of 84% [20]. In a retrospective study, Medina-Ornelas et al. compared [^68^Ga]Ga-PSMA-11 PET/CT to [^18^F]FDG PET/CT in 21 cases covering all breast cancer subtypes, including 5 cases of TNBC [21]. The study revealed PSMA positivity in 76% of primary breast cancer tumors and 90% of metastatic lesions in all breast cancer subtypes, although PSMA scans exhibited a lower detection rate than [^18^F]FDG in the analyzed group. Arslan et al. presented a comparison between PSMA immunohistochemical expression and PSMA radiotracer uptake in breast cancer, which indicated a positive and similar level [17]. Bertagna et al. presented a systematic review, which included 11 case studies evaluating radiolabeled PSMA PET/CT in breast cancer; all of them were ^68^Ga-PSMA imaging [18]. Initial results of a study assessing [^68^Ga]Ga-PSMA-11 PET/CT revealed that PSMA-targeted radioligand therapy may be considered as an innovative therapeutic strategy in patients with advanced TNBC [19].

In our study, we prospectively included patients with advanced-staged TNBC and performed a direct comparison between [^18^F]PSMA-1007 and [^18^F]FDG. The [^18^F]PSMA-1007 scans demonstrated in vivo the positive expression of PSMA in the primary tumor and metastases of TNBC. In general, PET/CT with [^18^F]PSMA-1007 showed high lesion detectability. In most patients, TNBC imaging with [^18^F]PSMA-1007 PET/CT was concordant with [^18^F]FDG PET/CT, with only some discrepancies in the level of local disease and lymph node metastases. In patients with distant metastases, however, a higher number of metastatic lesions was observed using [^18^F]PSMA-1007 than [^18^F]FDG. It was mostly attributed to a higher accumulation of ^18^F-PSMA in distant metastases (expressed by higher SUV_max_ values) than [^18^F]FDG. In addition, [^18^F]PSMA-1007 showed higher lesion detectability in the regions with high physiological [^18^F]FDG accumulation, i.e., in the brain and in the adjacent bones (cranium) (Figure 2B). On the other hand, a concern may arise that the high physiological uptake of ^18^F-PSMA-1007 in organs, like the liver, may limit the detectability of distant metastases. However, we cannot draw any general conclusions with regard to that since the only patient who had liver metastases in our study group did show ^18^F-PSMA-1007 uptake higher than the background, as depicted in Figure 4D.

Currently, the [^18^F]FDG PET examinations contribute significantly to breast cancer imaging, surpassing other imaging modalities, particularly in the detection of extra-axillary (infraclavicular, supraclavicular, and internal mammary) nodal metastases and occult distant metastases. However, as shown by our results, ^18^F-PSMA may improve staging accuracy, especially in cases of widespread disease involving the brain, bone, and visceral organs.

By demonstrating the expression of PSMA in vivo in TNBC lesions, the results of our study support the use of PSMA ligands for imaging, in particular in the context of a potential therapy with PSMA ligands labeled with beta-emitting radionuclides that might be an attractive option for advanced-stage TNBC patients. The overexpression of PSMA has already been validated in vitro via immunohistochemistry, revealing significantly higher PSMA expression in metastatic lesions [10,11,22]. Numerous authors underline the significance of neoangiogenesis in the development of distant metastases [23]. To date, several studies have linked PSMA to cancer-related angiogenesis, emphasizing its overexpression in tumor-associated vasculature, which explains the highest uptake of [^18^F]PSMA-1007 in metastases that we observed in our study [10]. Therefore, PSMA is gaining considerable attention as a new vascular target in advanced breast cancer patients. The substantial uptake of [^18^F]PSMA-1007 in PET/CT, particularly in distant metastases, opens up new possibilities for radionuclide therapy in advanced-stage TNBC patients.

Limitations of our study include its single-center design and a small number of patients. As no previous experiences with a direct comparison of [^18^F]PSMA-1007 and [^18^F]FDG in TNBC are available in the literature, the presented paper should be regarded as a pilot study. Its main purpose was to obtain a preliminary comparative analysis of both imaging modalities in a small number of patients. We also aimed at the inclusion of patients with different clinical scenarios, i.e., not only in case of relapse but also during neoadjuvant chemotherapy (CTH) and also prior to surgery. This explains the apparent heterogeneity of the study group. In spite of that, the results of the study show that [^18^F]PSMA-1007 PET/CT can be theoretically used in all these clinical indications. We have to keep in mind, however, that [^18^F]PSMA-1007 PET/CT will probably not substitute [^18^F]FDG PET/CT in the diagnosis of TNBC. First of all, [^18^F]FDG shows satisfactory diagnostic accuracy [24,25,26]. Secondly, [^18^F]FDG is widely available and cheaper than ^68^Ga- or ^18^F-labeled PSMA ligands. The potential role of PSMA ligands in this clinical setting is for in vivo evaluation of PSMA expression and, subsequently, assessment of potential indications to targeted radionuclide therapy using PSMA ligands labeled with beta-emitters (^177^Lu) or, in the future, also alpha-emitters (e.g., ^225^Ac) as it is already the case in the treatment of prostate cancer [27]. This theranostic approach is also well-established in the case of the somatostatin analogs used in the diagnosis and therapy of neuroendocrine tumors [28].

Another limitation is the absence of in vitro confirmation. First, no biopsy of the lesions was performed to confirm the metastatic character of the detected lesions in lymph nodes or organs. Subsequently, material for immunohistochemical evaluation of PSMA expression in the detected lesions was not available. From the clinical point of view, histopathological verification was deemed unnecessary, as the patients were already being treated for advanced-stage cancer, and a biopsy would not have altered the management. Therefore, conducting an invasive procedure to assess PSMA expression in vitro solely for research purposes would have been ethically unacceptable. It could also be argued that both PET/CT procedures were not performed simultaneously but with some interval in between (up to 30 days). This should be explained with logistic issues as [^18^F]PSMA-1007 production is not performed on an everyday basis by our supplier.

Finally, it should be noted that research on prostate cancer imaging has been conducted mostly using a few ^68^Ga-labeled PSMA ligands [29]. Similarly, in previous studies of breast cancer patients, ^68^Ga-labeled compounds were used [16,20,21]. In the meantime, ^18^F-labeled PSMA ligands for PET imaging have been developed, such as [^18^F]DCFPyL ([^18^F]piflufolastat), [^18^F]PSMA-1007, [^18^F]rhPSMA-7 with advantages and disadvantages related primarily to the physical and chemical properties of the radioisotope [30]. The difference between the most common tracers, [^68^Ga]Ga-PSMA-11 and [^18^F]PSMA-1007, is mostly technical, as ^68^Ga can be produced on-site if a center possesses a specific generator, whereas ^18^F production requires a cyclotron. In prostate cancer patients, PET/CT scans performed with both radiopharmaceuticals are of comparable diagnostic accuracy, although some studies indicated a higher number of unspecific findings in the case of [^18^F]PSMA-1007 [31,32]. There are no data on the differences between these modalities in breast cancer patients, but it should be presumed that both examinations should provide comparable results.

## 5. Conclusions

To the best of our knowledge, this is the first prospective study directly comparing [^18^F]PSMA-1007 and [^18^F]FDG PET/CT in TNBC. This preliminary study performed on a limited number of patients showed good performance of [^18^F]PSMA-1007 PET/CT in the detection of TNBC lesions. [^18^F]PSMA-1007 showed high accumulation in distant metastases, higher than in the standard PET/CT imaging using [^18^F]FDG, and, consequently, it presented some advantages with regard to the detection of distant metastases, in particular in the brain. The findings require further evaluation as the potential use of PSMA-based radiopharmaceuticals may not only improve diagnostic imaging of TNBC but also potentially lead to the introduction of novel therapeutic options.

## Data Availability

The data can be shared up on request.

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
