# Peer review of "Head-to-Head Comparison of [^18^F]PSMA-1007 and [^18^F]FDG PET/CT in Patients with Triple-Negative Breast Cancer"

_cancers, 2024, doi:10.3390/cancers16030667_

Round 1

Reviewer 1 Report

Comments and Suggestions for Authors

The present manuscript is a preliminary experience of PSMA in breast cancer patients with triple negative disease. 

Interesting paper, very limited number of patients, but with some potential impact on the clinical practice. Please below some minor comments:

1- please use the terms neoangiogenesis rather than neovascularization

2- Why the authors used only 60 min after the injcetion of the tracer? Usually the time for 18F-PSMA-1007 is 90-120 min?

3- Please use the internationa nomenclature for the tracers

4- TBR would be preferable than SUVmax.

5- The setting of disease would  benefit from PSMA rather than FDG would be discussed better in the discussion. For exmaple in the iniatial staging, FDG can provide prognostic information, while PSMA? Probably only in the widespread disease, with brain, bone and visceral disease, it can help!

6- the high liver accumulation of PSMA can reduce the detectability of the disease, please explain how to overpass this limitation. 

Author Response

The present manuscript is a preliminary experience of PSMA in breast cancer patients with triple negative disease. Interesting paper, very limited number of patients, but with some potential impact on the clinical practice. Please below some minor comments:

Dear Reviewer, Thank you for your comments and valuable advice. Here is a point-by-point response to your feedback. All revisions in the manuscript are highlighted in the attached file.

1- please use the terms neoangiogenesis rather than neovascularization

Thank you for bringing this to our attention. We have duly noted your suggestion and have updated the term "neovascularization" to "neoangiogenesis" throughout the manuscript.

2- Why the authors used only 60 min after the injcetion of the tracer? Usually the time for 18F-PSMA-1007 is 90-120 min?

Thank you for your feedback. We acknowledge your comment, and we want to clarify that we conducted the examinations 60 minutes post-tracer injection based on the supplier's recommendation. This time point was chosen for standardization across all 10 patients.

3- Please use the internationa nomenclature for the tracers

Thank you for your advice. We have corrected the manuscript and used international nomenclature: [18F]FDG and [18F]PSMA-1007.

4- TBR would be preferable than SUVmax.

Thank you for your recommendation. Following your suggestion, we re-evaluated all patients and measured TBR. The updated data has been included in Table 2 and in the text. We have changed the Table 2 for better clarity of data. We appreciate your valuable input, and we believe this addition enhances the clarity and completeness of our findings.

5- The setting of disease would  benefit from PSMA rather than FDG would be discussed better in the discussion. For exmaple in the iniatial staging, FDG can provide prognostic information, while PSMA? Probably only in the widespread disease, with brain, bone and visceral disease, it can help!

Thank you for your valuable feedback. We appreciate your suggestion regarding the discussion. In response, we have expanded our discussion to provide a more comprehensive analysis. Specifically, we have highlighted that while FDG can offer prognostic information in initial staging, PSMA may prove particularly beneficial in cases of widespread disease involving the brain, bone, and visceral areas. We hope these additions address your concerns and enhance the overall clarity of our paper.

6- the high liver accumulation of PSMA can reduce the detectability of the disease, please explain how to overpass this limitation.

Thank you for your comment. We were also concerned that the high liver uptake of 18F-PSMA-1007 may limit the detectability of liver metastases. However, we cannot draw any general conclusions with regard to that since the only patient who had liver metastases in our study group did show 18F-PSMA-1007 uptake higher than background as depicted in Fig. 4 D. This limitation can be potentially mitigated by delayed scans of the liver region. We added some comments on that in the discussion.

Reviewer 2 Report

Comments and Suggestions for Authors

The Manuscript entitled: “ Head-to-Head Comparison of 18F-PSMA-1007 and 18F-FDG 2 PET/CT in Patients with Triple Negative Breast Cancer”, has a good scientific sound for the reader of the Cancers journal. I conclude that the paper should undergo minor revisions, and after correction, it is suited for further procedure of publication. My comments are as follows:

1-      In the Introduction section, the manuscript lacks a comparison of methods used in this work with other relevant works.

2-      There is no information regarding SUVs in the introduction section.

3-      In the Materials and Methods section, PET/CT examination, there is no information about CT imaging protocols (such as slice thickness, ……. ).

4-      The authors should include some words in the text, which they used in the tables and figures such as CTH, RTH, and CPA.

5-      I think the limited number of investigated patients (10 patients) is an important issue, which is needed to be clarified.

6-      In the discussion section, some sentences regarding the current and possible future clinical usage of these two methods are required.

7-      Table 1 is not informative. Please provide a self-explanatory table understandable without needing to refer to the text.

8-      Some grammar and spelling in the English language needs to correction.

Comments on the Quality of English Language

  Some grammar and spelling in the English language needs to correction.

Author Response

The Manuscript entitled: “ Head-to-Head Comparison of 18F-PSMA-1007 and 18F-FDG 2 PET/CT in Patients with Triple Negative Breast Cancer”, has a good scientific sound for the reader of the Cancers journal. I conclude that the paper should undergo minor revisions, and after correction, it is suited for further procedure of publication. My comments are as follows:

Dear Reviewer, Thank you for your comments and valuable advice. Here is a point-by-point response to your feedback. All revisions in the manuscript are highlighted in the attached file for your convenience.

1-      In the Introduction section, the manuscript lacks a comparison of methods used in this work with other relevant works.

Thank you for your valuable feedback. While you suggest an extension of the introduction, another reviewer has recommended shortening it. We would like to emphasize that a comprehensive comparison of the methods has been included in the Discussion section of our manuscript. To better address your request, could you please provide additional information regarding the specific methods you are interested in? Are you referring to FDG, other PET tracers, or perhaps MRI methods? This clarification will help us tailor our response to meet your expectations more effectively.

2-      There is no information regarding SUVs in the introduction section.

I appreciate your feedback. Could you please provide more details or specify which aspects of SUVmax you believe should be included in the introduction section? This will help us address your comment more accurately. 

3-      In the Materials and Methods section, PET/CT examination, there is no information about CT imaging protocols (such as slice thickness, ……. ).

Both PET images were fused with the CT image obtained using the following parameters: 1.25-mm layer thickness, 1.375:1 pitch, 50-cm DFOV 50 and 512 × 512 matrix. We have added these details to the Methods section.

4-      The authors should include some words in the text, which they used in the tables and figures such as CTH, RTH, and CPA.

Thank you for your comment. We included more abrevations into the text.

5-      I think the limited number of investigated patients (10 patients) is an important issue, which is needed to be clarified.

Thank you for bringing up the concern about the limited number of investigated patients (10 patients). The constrained sample size is primarily attributed to the associated costs of the examinations. Additionally, the study faced challenges in identifying an adequate number of patients with the specific type of cancer at an advanced stage who were willing to undergo both examinations. Unfortunately, some patients were too weak to participate in two separate scans. Despite these limitations, we believe the insights gained from this cohort provide valuable initial observations. If you have further suggestions or inquiries, please feel free to share them.

6-      In the discussion section, some sentences regarding the current and possible future clinical usage of these two methods are required.

Thank you for your input. We appreciate your suggestion, and we have incorporated information regarding the current and potential future clinical usage of both imaging modalities in the discussion section. We hope these additions enhance the overall understanding of the implications of our findings. If you have any further recommendations or specific areas you would like us to address, please let us know. We value your feedback.

7-      Table 1 is not informative. Please provide a self-explanatory table understandable without needing to refer to the text.

We found Table 1 to be necessary to provide basic clinical history of each of the patients. In the text, there is only a brief summary of the entire group without looking into the details of each individual case. Table 1 should demonstrate the heterogeneous character of the group and should be regarded as a supplement for a clinically oriented reader.

8-      Some grammar and spelling in the English language needs to correction.

Thank you for your feedback. We have found some errors and corrected them.

Reviewer 3 Report

Comments and Suggestions for Authors

This study evaluated the difference between FDG and PSMA(18F-PSMA-1007) uptake in triple-negative breast cancer patients. The author concluded that 18F-PSMA PET/CT will enhance both diagnostic and therapeutic approaches for the aggressive breast cancer subtype.

Please shorten the introduction section. 

Were the PET images interpreted blind to clinical information?

What was the standard reference to confirm the true positive or negative in this study?

The author should add the evaluation of the PET uptake with tumor/blood or background ratio. Considering the physiological uptake pattern of PSMA, the high physiological uptake of PSMA in the liver made it difficult to identify the liver metastasis. It is not easy to identify the liver lesion like Figure 3 without referencing other images.

Define "CTH" for clarity.

In the case of the obtained PET scan to evaluate the treatment effect, FDG and PSMA uptake might be affected by the type of therapy. Please discuss the possible effect of therapy on PET uptakes.

Please add the representative false positive PET uptake observed in the interpretation by the specialist.

[18F]PSMA-1007 is predominantly excreted from the hepatobiliary tract and has low renal excretion. However recent reports showed high urinary excretion might be occasionally found in PSMA-1007. Guessing from MIP images in this study, bladder uptake varied in each case. Was it prominent in the patient for the case just receiving chemotherapy in this study?

Comments on the Quality of English Language

Minor editing of English language required

Author Response

This study evaluated the difference between FDG and PSMA(18F-PSMA-1007) uptake in triple-negative breast cancer patients. The author concluded that 18F-PSMA PET/CT will enhance both diagnostic and therapeutic approaches for the aggressive breast cancer subtype.

Dear Reviewer, Thank you for your comments and valuable advice. Here is a point-by-point response to your feedback. All revisions in the manuscript are highlighted  in attached file for your convenience.

Please shorten the introduction section.

Thank you for your feedback. Could you please specify which particular section of the introduction you would like us to shorten? It's important to note that other reviewers have suggested expanding the introduction, so any guidance on the specific section to be shortened will help us address your comment more effectively.

Were the PET images interpreted blind to clinical information?

The PET images were not interpreted blindly to clinical information. This decision was made because various disease and patient conditions can influence radiotracer uptake. Therefore, each specialist requires detailed information about the patient's overall condition and medical history, not limited to oncological but also encompassing other diseases, surgeries and medications.

What was the standard reference to confirm the true positive or negative in this study?

In a few cases, the detected lesions were verified histopathologically (as the second focus of TNBC in the same breast). In the majority of cases, however, disseminated lesions were only confirmed with follow-up imaging using CT or MR. A surgical intervention was ethically not justified and contraindicated from clinical point of view.

The author should add the evaluation of the PET uptake with tumor/blood or background ratio. Considering the physiological uptake pattern of PSMA, the high physiological uptake of PSMA in the liver made it difficult to identify the liver metastasis. It is not easy to identify the liver lesion like Figure 3 without referencing other images.

Thank you for your comment. We were also concerned that the high liver uptake of 18F-PSMA-1007 may limit the detectability of liver metastases. However, we cannot draw any general conclusions with regard to that since the only patient who had liver metastases in our study group did show 18F-PSMA-1007 uptake higher than background as depicted in Fig. 4 D. This limitation can be potentially mitigated by delayed scans of the liver region. We added some comments on that in the discussion.

Define "CTH" for clarity.

The abbreviation "CTH" means “chemotherapy” and it is defined in the text and in the table for clarity

In the case of the obtained PET scan to evaluate the treatment effect, FDG and PSMA uptake might be affected by the type of therapy. Please discuss the possible effect of therapy on PET uptakes.

Please add the representative false positive PET uptake observed in the interpretation by the specialist.

Thank you very much for pointing out the need of including patient’s treatment history into the interpretation of PET/CT images. Both, 18F-FDG and 18F-PSMA-1007 uptake may be locally elevated for several weeks following surgery or radiation therapy. In our cohort, surgical and radiation therapy was performed in a few patients only and it was a relatively long time before recruitment to the study. So, we did not find any case demonstrating false positive lesions related to therapy. On the other hand, current chemotherapy could decrease uptake of the radiopharmaceuticals in malignant lesions leading to the occurrence of false negative findings. We cannot exclude such a situation as we could only refer to the lesions that we were aware of. False negative scans would decrease the method's sensitivity. In our study, we did not calculate sensitivity of both PET/CT scans due to the low number of subjects, heterogenous clinical background etc.

[18F]PSMA-1007 is predominantly excreted from the hepatobiliary tract and has low renal excretion. However recent reports showed high urinary excretion might be occasionally found in PSMA-1007. Guessing from MIP images in this study, bladder uptake varied in each case. Was it prominent in the patient for the case just receiving chemotherapy in this study?

Thank you for your comment. We thoroughly examined the data, and unfortunately, we did not identify any correlations in the case of patients who had undergone chemotherapy in our study.

Reviewer 4 Report

Comments and Suggestions for Authors

This research study is focused on exploring a promising avenue for improving the diagnosis and treatment of triple-negative breast cancer (TNBC), which is a highly aggressive form of breast cancer that has limited therapeutic options. The researchers are primarily investigating the prostate-specific membrane antigen (PSMA), which is known to be present in prostate cancer but has also been identified in breast cancer. The study is using 18F-PSMA-1007 PET/CT to assess the PSMA in vivo expression in TNBC patients and compare it with the standard 18F-FDG-PET/CT method. The aim of this study is to determine whether 18F-PSMA PET/CT can outperform the current methods in detecting distant metastases, particularly in the brain. The researchers have found that the 18F-PSMA PET/CT method may be more effective in detecting distant metastases compared to the standard 18F-FDG-PET/CT method. This finding is significant because it suggests that the new method may be more accurate in diagnosing and treating TNBC. The research methodology is well defined. However, the study needs some minor revisions before it can be considered conclusive. Firstly, the number of references provided in the study is not sufficient, and the authors should include recent studies to elaborate on their points. Additionally, the authors should include SUV values in Table format of each patient for both tracers 18F-FDG-PET/CT and 18F-PSMA PET/CT with the interest of target organs. These revisions will help to strengthen the study and provide more detailed insights into the effectiveness of the new method.

Comments on the Quality of English Language

Moderate editing of English language required.

Author Response

This research study is focused on exploring a promising avenue for improving the diagnosis and treatment of triple-negative breast cancer (TNBC), which is a highly aggressive form of breast cancer that has limited therapeutic options. The researchers are primarily investigating the prostate-specific membrane antigen (PSMA), which is known to be present in prostate cancer but has also been identified in breast cancer. The study is using 18F-PSMA-1007 PET/CT to assess the PSMA in vivo expression in TNBC patients and compare it with the standard 18F-FDG-PET/CT method. The aim of this study is to determine whether 18F-PSMA PET/CT can outperform the current methods in detecting distant metastases, particularly in the brain. The researchers have found that the 18F-PSMA PET/CT method may be more effective in detecting distant metastases compared to the standard 18F-FDG-PET/CT method. This finding is significant because it suggests that the new method may be more accurate in diagnosing and treating TNBC. The research methodology is well defined. However, the study needs some minor revisions before it can be considered conclusive.

Dear Reviewer, Thank you for your comments and valuable advices. Here is a point-by-point response to your feedback. All revisions in the manuscript are highlighted  in attached file for your convenience.

Firstly, the number of references provided in the study is not sufficient, and the authors should include recent studies to elaborate on their points.

Thank you for your insightful feedback. We have incorporated additional references as per your suggestion.

Additionally, the authors should include SUV values in Table format of each patient for both tracers 18F-FDG-PET/CT and 18F-PSMA PET/CT with the interest of target organs.

Thank you for your comment. We provided SUVmax and TBR values in Table 2, for both 18F-FDG and 18F-PSMA PET/CT examinations. We created new table for better clarity.

These revisions will help to strengthen the study and provide more detailed insights into the effectiveness of the new method.

Thank you for your comments. We hope these additions address your concerns and enhance the overall clarity of our paper.

Round 2

Reviewer 3 Report

Comments and Suggestions for Authors

None